# Relation3DMOT: Exploiting Deep Affinity for 3D Multi-Object Tracking from View Aggregation

**DOI:** 10.3390/s21062113

**Published:** 2021-03-17

**Authors:** Can Chen, Luca Zanotti Fragonara, Antonios Tsourdos

**Affiliations:** School of Aerospace, Transport and Manufacturing, Cranfield University, Bedford MK43 0AL, UK; l.zanottifragonara@cranfield.ac.uk (L.Z.F.); a.tsourdos@cranfield.ac.uk (A.T.)

**Keywords:** 3D multi-object tracking, sensor fusion, deep affinity, relation learning, neural network

## Abstract

Autonomous systems need to localize and track surrounding objects in 3D space for safe motion planning. As a result, 3D multi-object tracking (MOT) plays a vital role in autonomous navigation. Most MOT methods use a tracking-by-detection pipeline, which includes both the object detection and data association tasks. However, many approaches detect objects in 2D RGB sequences for tracking, which lacks reliability when localizing objects in 3D space. Furthermore, it is still challenging to learn discriminative features for temporally consistent detection in different frames, and the affinity matrix is typically learned from independent object features without considering the feature interaction between detected objects in the different frames. To settle these problems, we first employ a joint feature extractor to fuse the appearance feature and the motion feature captured from 2D RGB images and 3D point clouds, and then we propose a novel convolutional operation, named RelationConv, to better exploit the correlation between each pair of objects in the adjacent frames and learn a deep affinity matrix for further data association. We finally provide extensive evaluation to reveal that our proposed model achieves state-of-the-art performance on the KITTI tracking benchmark.

## 1. Introduction

Multi-object tracking in a 3D environment (3D MOT) plays a crucial role in the environmental perception of autonomous systems [1,2,3,4]. Typically, the 3D MOT task first localizes all the surrounding 3D objects in a sequence, and then assigns them the consistent identity numbers (IDs). As a result, the same objects in the sequence should be given the same IDs, which is then used to predict the trajectories of the autonomous systems surrounding objects and can be used to produce safe path planning for autonomous navigation (i.e., obstacle avoidance). In recent years, 2D multi-object tracking has made great progress, especially in the computer vision domain [5,6,7,8,9]. However, camera sensors are unlikely to provide depth information, unless relatively computationally expensive methods such as stereo-vision are used. Furthermore, cameras are typically quite sensitive to lighting and environmental conditions (e.g., overexposure and day/night cycles). As a result, some researchers draw attention to 3D MOT using point cloud data, especially when *PointNets* [10,11] allow directly dealing with the irregular point clouds in an efficient way. *AB3DMOT* [4] directly applies the Kalman filter [12] and Hungarian algorithm [13] over the obtained 3D detections to achieves real-time performance. However, the point clouds are incapable of offering rich texture information, which makes the Lidar-only detectors struggle to capture high-resolution features from the point clouds, especially for the objects in the long-distance that are likely to just contain few points. Besides, the hand-crafted approaches Kalman filter [12] and Hungarian algorithm [13] only include low-level state features of the objects (e.g., location, velocity, and acceleration) for the state update and data association, which are not robust compared to the deep learning methods that could extract the object states to the high-level features with contextual information for the data association.

In order to improve the reliability and safety, recent tracking-by-detection approaches [2,14,15] first combine the camera with the Lidar sensor that is capable of offering precise spatial information. Leveraging the sensor fusion technology and redundant information from multiple sensors, the performance of 3D MOT can be significantly boosted. After that, a pairwise feature similarity between any two objects in the different frames is learned from fused features. For example, *GNN3DMOT* [14] captures the 2D/3D appearance features and 2D/3D motion features from the image and the point cloud data. However, the model just projects the 3D detections in the point clouds to the images to obtain the 2D information, which leads to lower precision and recall of the performance. With regard to the data association problem, *JRMOT* [15] employs Joint Probabilistic Data Association (JPDA) [16] to assign different IDs to the corresponding objects, which is unlikely to include semantic features for the object states. *mmMOT* [2] learns an affinity matrix from the 2D/3D appearance features directly for data association. However, the motion features and occlusion conditions are not involved in the model.

It is easy to observe that capturing the discriminative feature, which is beneficial for distinguishing different objects, is the critical process when learning the affinity matrix and the following data association. One efficient method to solve this problem is to represent the objects in the adjacent frames as a directed graph. Specifically, each object can be treated as a node in the graph, and the relationship between an object pair is the edge between related two nodes. As a result, the problem of exploiting the discriminative feature between two objects is convert to learn the relations between the nodes in a directed graph. Consequently, we propose a deep neural network by employing the graph neural network into the 3D MOT to better exploit discriminative features.

Furthermore, considering the fact that the nodes in the graph are not structured and unordered, we are unlikely to leverage the advantages of convolutional neural networks (CNNs) to exploit the features. Prior works for both 2D MOT and 3D MOT [2,3,7,8,14] use a multi-layer perceptron (MLP) to capture contextual features for each node. However, it is not efficient to learn local features between nodes using a MLP, as the MLP operation is not a spatial convolution. As a result, we introduce a novel convolutional operation, named RelationConv, to better exploit relations between nodes in the graph.

In order to learn temporal-spatial features for the objects in the sequence, we also propose a feature extractor that jointly learns appearance features and motion features from both 2D images and 3D point clouds. Specifically, we use the off-the-shelf 2D image and 3D point cloud detectors to extract respective features for the appearance features of the objects. We learn the motion features by building a subnetwork that just takes the 2D bounding boxes in the images as input. We finally fuse the appearance feature and motion feature together for further data association. We summarize our contributions as below.
We represent the detected objects as the nodes in a directed acyclic graph and propose a graph neural network to exploit the discriminative features of the objects in the adjacent frames for 3D MOT. Specifically, a directed graph is constructed by treating each object feature as the node. Consequently, the Graph Neural Network (GNN) technique could be employed to update node feature by combining the features of other nodes.We propose a novel joint feature extractor to learn both 2D/3D appearance features and 2D motion features from the images and the point clouds in the sequence. In particular, the 2D/3D appearance features are learned from the image and the point cloud respectively by applying corresponding feature extractors. The motion model uses the parameters of the 2D bounding box as the motion cues to capture the motion features, which are finally aggregated with the 2D/3D appearance features to obtain the joint feature.We propose the RelationConv operation to efficiently learn the correlation between each pair of objects for the affinity matrix. The proposed operation is similar to the kernel of Convolutional Neural Network (CNN) that is used on the standard grid data, but our operation is more flexible and can be applied over the irregular data (e.g., graphs).

## 2. Related Work

### 2.1. 2D Multi-Object Tracking

MOT methods can be categorized into online and offline methods. Online method envisage the prediction of data association between the detection in current frame and a few past frames. Online are the standard for real-time applications. Early 2D trackers [17,18,19,20,21] enhance the robustness and accuracy of tracking by exploiting and combining deep features in the different layers. However, these integrated features from multiple layers are not helpful when the targets are heavily occluded or even unseen in a certain frame. Several 2D MOT methods [22,23,24] employ correlation filters to improve the decision models. Deep reinforcement learning is used in [25] to efficiently predict the location and the size for the targets in the new frame.

On the other hand, offline methods aim at finding the solution as an optimization problem for the whole data sequence. Some models [26,27] build a neural network with a min-cost flow algorithm to optimize the total cost for the data association problem.

### 2.2. 3D Multi-Object Tracking

3D object detection has achieved great success in the recent years, especially given PointNets [10,11] are capable of directly processing the unstructured point cloud data in an efficient way. As a result, many researchers draw attention to 3D object tracking based on accurate 3D detection results. Some approaches [28,29,30] first predict the 3D objects from off-the-shelf 3D detectors, followed by a filter-based model to track the 3D objects continuously. *mmMOT* [2] builds an end-to-end neural network to extract features for the detected objects and data association. Specifically, it employs a 2D feature extractor [31] and a 3D feature extractor [10] to capture the 2D and 3D features of objects in the adjacent two frames. A sensor fusion module is then proposed to aggregate multimodality features, which is then used for data association. However, the model only learns appearance features for the detected objects, and motion features are not considered. Alternatively, *GNN3DMOT* [14] proposes a joint feature extractor to learn discriminating appearance features for the objects from the images and the point clouds, and then employs an LSTM neural network to capture the motion information. Finally, a batch triplet loss is processed for data association.

### 2.3. Joint Multi-Object Detection and Tracking

Joint multi-object detection and tracking method has become popular as it might lead to a sub-optimal solution if the detection task and the tracking task are decoupled. Recent methods [6,32] construct an end-to-end framework with a multi-task loss to directly localize the objects and associate them with the objects in previous frame. Similarly, *FaF* [33] first converts the sequential point clouds into stacked voxels, and then applies standard 3D CNN over 3D space and time to predict 3D objects location, associate them in the multiple frames, and forecast their motions.

### 2.4. Data Association in MOT

Data association is an essential problem in the MOT task to assign the same identity for the same objects in the sequence. Traditionally, the Hungarian algorithm [13] minimizes the total cost of the similarity for each pair of observations and hypotheses. Conversely, *JPDA* [16] considers all the possible assignment hypotheses and uses a joint probabilistic score to associate objects in the different frames. Modern works [26,34] first represent the objects and corresponding relations as a directed acyclic graph. Each object is treated as a node in the graph, and the relation between each pair of objects represents a graph edge. Successively, the data association problem can be solved as a linear programming problem to find the optimal solution.

## 3. Model Structure

Our proposed 3D MOT network follows the tracking-by-detection paradigm. As shown in Figure 1, the framework takes a sequence of images and related point clouds as input, and consists of three modules: (a) a *joint feature extractor* to capture the appearance feature and the motion feature from the 2D images and 3D point clouds, (b) a *feature interaction module* that takes the joint feature as input and uses proposed RelationConv operation to exploit the correlation between the pairs of objects in the different frames, (c) a *confidence estimator* predicts if a certain detected object is a valid detection, and (d) a *data association module* to compute the affinity matrix for associating the objects in the adjacent frames.

### 3.1. Problem Statement

As an online 3D MOT method, our model performs objects association in every two consecutive frames from a given sequence. Assume that a new identity number will be assigned if an object reappears in the frame due to the occlusion or misdetection conditions. Consequently, our model focuses on correlation learning without considering the re-establishment of the associations for the objects across the temporal gaps. We refer to the current frame at time *t* with *N* detected objects as $X^{t}=\left\{{x_{i}}^{t}|i=1,2,\ldots ,N\right\}$, and to the previous frame at time t−1 with *M* detected objects as $X^{t-1}=\left\{{x_{j}}^{t-1}|j=1,2,\ldots ,M\right\}$. We aims at exploiting the discriminative feature for each ${x_{i}}^{t}$ and ${x_{j}}^{t-1}$ pair, predicting a feature affinity matrix for the correct matching, and finally assigning the matched IDs to the corresponding objects in the current frame *t*.

### 3.2. 2D Detector

For what concerns the object detection, there are two main methods to localize the objects in the frames. One method is to use existing, state-of-the-art, 3D detectors (e.g., *PointRCNN* [35] and *PointPillar* [36]) to predict 3D bounding boxes, which are then projected to corresponding images to obtain 2D bounding boxes. However, we believe that the fact that the point clouds are lacking of rich color and texture information, so that this will not provide sufficient semantic information to the detected objects. As a result, it is still not efficient to localize the objects precisely. Therefore, another alternative method is to use off-the-shelf 2D detectors to predict 2D bounding boxes. With respect to the 3D location, we employ the method from the work in [37] to estimate the localization of the objects. We use this method and introduce *RRC-Net* [38] as our 2D detector model, as we observed that it should obtain the highest recall and accuracy.

### 3.3. Joint Feature Extractor

In order to exploit sufficient information for the detected objects in the frames, we propose a joint feature extractor (see Figure 1a) to learn deep representations from both the images and the point clouds. Specifically, we employ a modified *VGG-16* [31] to extract 2D appearance features, and then apply *PointNet* [10] over the trimmed points, which are obtained by extruding related 2D bounding boxes into the 3D frustums in the point cloud, to capture 3D appearance features and predict 3D bounding boxes. After that, we propose a sensor fusion module to aggregate the 2D and 3D appearance features together for further feature interaction. With regard to the motion features, we build a subnetwork to learn the high-level motions of objects just using the information of the 2D bounding boxes.

#### 3.3.1. 2D Appearance Feature Extraction

As shown in Figure 1a, we take the objects in the 2D bounding boxes in the image as input, which are then cropped and resized to the fixed size 224×224 to adapt them to the *VGG-16* [31] feature extractor. The work in [18] indicates that the features in the different CNN layers contain different semantic properties. For example, a lower convolutional layer is likely to capture more detailed spatial information than low-level features, and a deeper convolutional layer would tend to capture more abstract and high-level information. As a result, inspired by the works in [2,18], we embed a skip-pool [39] method into the *VGG-16* network (see Figure 2) to involve all the features at the different levels for the global feature generation, which is then treated as the 2D appearance feature for the corresponding detected object.

#### 3.3.2. 3D Appearance Feature Extraction

We first obtain the 3D points of each object by extruding the corresponding 2D bounding box into a 3D bounding frustum, where the 3D points on the object are then trimmed from. After that, we apply *PointNet* [10] to capture the spatial features to predict the corresponding 3D bounding box.

#### 3.3.3. Motion Feature

Our model is a purely local tracker that only associates the objects in the adjacent frames through time without considering the knowledge of previous and future frames. As a result, the motion feature is designed to independently estimate the relative spatial relationship for each pair of objects in the different frames. Unlike traditional motion models (e.g., Kalman filter [12] and Long Short-Term Memory (LSTM) [40]), we learn the relative spatial displacement directly from the location of the 2D/3D bounding box. However, considering the fact that we observe the 3D appearance features already containing the spatial information for the foreground 3D points, it is not necessary to embed the 3D bounding boxes into the motion cues. Consequently, we only use the 2D bounding boxes as the motion cues. After aggregation with the 3D appearance that contains the 3D spatial information, we finally predict 2D/3D displacements of each object pair for spatial relationship learning.

With respect to our motion feature for each object, we define the 2D bounding box of a certain object as $B_{2d}=\left[x_{2d},y_{2d},w_{2d},h_{2d}\right]$, where $\left[x_{2d},y_{2d}\right]$ is the center of the 2D bounding box in the image, and $w_{2d},h_{2d}$ are the width and the height of the 2D bounding box, respectively. As a result, the motion cue is defined as $B=[B_{2d}]$ (see Figure 1a), which is then fed to a 3-layer MLP sub-network to capture the motion feature for each detected object.

#### 3.3.4. Features Aggregation and Fusion

We first aggregate 2D and 3D appearance features to have sufficient semantic features to exploit (e.g., spatial and color features) for each detected object. We propose three fusion operators: (1) an intuitive method is to add 2D and 3D appearance features together after forcing them to have the same feature channels; (2) another common approach is to concatenate 2D appearance feature and 3D appearance feature, after which a 1-layer MLP is used to adapt the dimension of the fused features; and (3) the last operator is an attention-based weighted sum. Specifically, as defined in Equation (Equation 1), a scalar score is first computed by learning a 1-dimensional feature from the features in the different sensors, after which the score is normalized by applying a sigmoid function. Finally, the weighted sum for the fusion is computed by using an element-wise multiplication operation on the normalized scores and corresponding features obtained from different sensors as defined in Equation (Equation 2). We compare the performance by applying different fusion operators on our model in the experiments and using a concatenation operator as our 2D and 3D appearance features aggregation method in our model.
(1)$$\alpha^{s}=\sigma \left(W_{a}^{s}⊗F^{s}\right)$$
where $W_{a}^{s}$ indicate the convolution parameters for the features $F^{s}$ obtained from a certain sensor *s*. $\alpha^{s}$ is the learned attention-weighted score for the sensor *s*. ⊗ is a multiplication operation.
(2)$$F^{\mathrm{fuse}}=\sum_{s=0}^{S}\left(\alpha^{s}⊙\left(W_{\mathrm{fuse}}^{s}⊗F^{s}\right)\right)$$
where $F^{\mathrm{fuse}}$ denote the features after fusion. $W_{\mathrm{fuse}}^{s}$ are the convolution parameters for the the features $F^{s}$ obtained from a certain sensor *s*. ⊙ is an element-wise multiplication.

We finally fuse the appearance feature and motion feature by concatenating the aggregated 2D/3D appearance feature with the related motion feature for further feature interaction.

### 3.4. Feature Interaction Module

After the joint feature extractor, we gain the fused features for *M* individual objects in the frame t−1 and *N* objects in the frame *t*. We then propose a feature interaction module, as shown in Figure 1b, to learn the relations between each pair of two objects (one object is in previous frame t−1 and the other object is in current frame *t*).

#### 3.4.1. Graph Construction

In order to efficiently represent the objects in the different frames, we treat the feature of each object as a node in a graph structure, and the edge between two nodes in the graph can indicate the relationship between two objects in the different frames. As a result, we first construct a directed acyclic graph structure to represent the objects in two adjacent frames as shown in Figure 3. Generally, a directed acyclic graph can be defined as G=(V,E), where $V\subset R^{F}$ are the nodes with *F* dimension, and E⊆V×V are edges connecting two nodes in the graph.

However, as we only learn the correlation between each object in the frame *t* and all the objects in the frame t−1, rather than taking the relations between the pair of objects in the same frame into account, our graph for the objects representation is constructed as $G=(V^{t},E^{t})$, $E^{t}\subseteq V^{t}\times V^{t-1}$, where $V^{t}$ indicate *N* objects at current frame *t*, and $V^{t-1}$, which are also the neighborhood nodes of $V^{t}$, denote *M* objects at previous frame t−1. The edge feature set $E^{t}$ for all the nodes in current frame are defined as Equation (Equation 3):(3)$$\begin{array}{cc} \; & X^{t}=\left\{x_{i}^{t}\in R^{F},i=1,2,\ldots ,N\right\}\\ \; & V^{t}=X^{t}\\ \; & {y_{i}}_{j}^{t}=\left|x_{i}^{t}-x_{j}^{t-1}\right|,j=1,2,\ldots ,M\\ \; & e_{i}^{t}=\left({y_{i}}_{1}^{t},{y_{i}}_{2}^{t},\ldots ,{{y_{i}}_{M}}^{t}\right),i=1,2,\ldots ,N\\ \; & E^{t}=\left\{e_{i}^{t}|i=1,2,\ldots ,N\right\} \end{array}$$
where $x_{i}^{t}$ indicates a certain object node in current frame *t*, ${y_{i}}_{j}^{t}$ denotes the edge feature connecting *i*-th object node $x_{i}^{t}$ in current frame *t* and *j*-th object node $x_{j}^{t-1}$ in previous frame t−1. The |·| is the absolute value operation. $e_{i}^{t}$ are the edge features connecting *i*-th object node in current frame *t* and all the *M* neighborhood nodes in previous frame t−1.

#### 3.4.2. Relation Convolution Operator

Considering irregular and unordered properties of the nodes in the graph, we are unlikely to use regular CNN filters (e.g., 3×3,7×7filters) on the unstructured graph for convolutional operation, as these filters are only suited to deal with regular grid data, such as images. As a consequence, the traditional method in the literature to deal with unstructured data can apply shared MLP on the graph to learn local and global contextual representations. However, it is not efficient to extract the spatially local features from unordered data using the shared 1×1 filters, which leads to small receptive fields. Inspired by the processing of standard convolutional kernels, *PointCNN* [41] learned a transformation matrix as a regular CNN filter to capture local features for the irregular point clouds, and it outperforms MLP-based methods *PointNets* [10,11] by a large margin.

In order to leverage the ability of CNNs, capable of extracting spatially-local correlation with large receptive fields, we propose a relation convolution operator, named RelationConv, to extract fine-grained local representations for the nodes on the graph. The advantages of our RelationConv are that it is similar to standard convolutional kernels and can work on irregular data (e.g., graphs). In standard CNNs, to obtain the abstract features $f_{X}$, the convolutional operation between the filters W and the feature map X can be defined as in Equation (Equation 4).
(4)$$f_{X}=W\;\cdot \;X$$
where X represents the feature map with standard grid distribution, and W are the convolutional filters. The operator “·” denotes element-wise multiplication.

Similarly, as defined in Equation (Equation 5), our RelationConv first learns a flexible filter W() by applying a shared MLP with a nonlinear function on the edge feature of the unordered graph, and then an element-wise multiplication operation is used on the learned filter W and the edge feature $E^{t}$ to extract the local feature for the nodes in the graph (see Figure 4). It is easy to observe that our flexible filter W is learned from all the nodes and corresponding neighborhood in the graph, which forces it to consider the global information in the graph, and also makes it independent to the ordering of the nodes.
(5)$$\begin{array}{cc} \; & W\left(E^{t}\right)=\mathrm{ReLU}\left(\mathrm{MLP}\left(E^{t}\right)\right)\\ \; & f_{E^{t}}=W\left(E^{t}\right)\;\cdot \;E^{t} \end{array}$$
where $E^{t}$ is the edge feature set. $W(E^{t})$ are the learnable and flexible filters obtained from a shared Multi-layer perceptron MLP() with a nonlinear function RELU() [42]. $f_{E^{t}}$ are the abstract features captured from our RelationConv operation.

#### 3.4.3. Feature Interaction

It is likely that the same objects in different frames should learn similar discriminative features. Therefore, the feature similarity should be dependent. In contrast, the feature similarity of two different objects should be low. As a result, the discriminative feature is beneficial for avoiding confusion while matching the objects.

Given the obtained fused features containing 2D/3D appearance features and 2D motion features, we propose a feature interaction module to learn the correlation between each object pair in the different frames as shown in Figure 1b. Rather than directly learning the deep affinity for further data association, we firstly employ a feature interaction module equipped with a 1-layer RelationConv network to learn more discriminative features, which allows the feature communication between two objects in adjacent frames before learning the affinity matrix.

#### 3.4.4. Confidence Estimator

As shown in Figure 1c, we build a 3-layer MLP binary classification network to predict the scalar scores $s^{\det}$ by applying the sigmoid() function to the final layer of the network defined as (Equation 6). The outputs are used as the validation scores that indicate the confidence of the true positive objects.
(6)$$s^{\det}=\frac{1}{1+\exp\left(-f_{\det}\right)}$$
where $f_{\det}$ indicate the last-layer features of the detected objects.

### 3.5. Data Association

#### 3.5.1. Affinity Matrix Learning

In order to associate and match the objects in the different frames, given the correlation representation after feature communication among objects, we use a 3-layer MLP subnetwork to learn an affinity matrix with a 1-dimension output feature $s^{A}$. The affinity matrix is capable of determining whether a certain object pair indicates a link. Besides, the scalar score in the matrix shows the confidence of the object pair associating with the same identity.

Furthermore, inspired by the work in [2], we learn a start-estimator and an end-estimator to predict whether an object is linked. Specifically, the start-estimator learns the scalar scores $s^{\mathrm{start}}$ to determine whether a certain object just appears in previous frame t−1. On the other hand, the end-estimator predicts the score $s^{\mathrm{end}}$ whether a certain object is likely to disappear in the next frame *t* (e.g., due to the presence of any hard occlusion or going out-of-bounds, etc.). The start-estimator and the end-estimator use an average pooling over the deep correlation representation to summarize the relations, and then employ the respective MLP network to learn the scalar scores for all the objects.

#### 3.5.2. Linear Programming

We obtain several binary variables for prediction scores from our proposed neural network as shown in Figure 1d. In summary, the detection score ${s_{i}}^{det}$ indicates the confidence whether the *i*-th object is a true positive detection. ${{s_{i}}_{j}}^{A}$, obtained from a softmax() function as defined in Equation (Equation 7), denotes the affinity confidence whether the *j*-th object in previous frame t−1 and the *i*-th object in current frame *t* are the same objects. ${s_{j}}^{start}$ denotes the confidence whether *j*-th object in previous frame t−1 starts a new trajectory in frame t−1. ${s_{i}}^{end}$ denotes the confidence whether *i*-th object in current frame *t* ends a trajectory in the frame *t*. Similar to ${s_{i}}^{det}$ obtained from a sigmoid() function in Equation (Equation 6), ${s_{j}}^{start},\;{s_{i}}^{end}$ are defined as Equation (Equation 8).
(7)$${{s_{i}}_{j}}^{A}=\frac{\exp(f_{ij})}{\sum_{k=0}^{N_{t-1}}\exp\left(f_{ik}\right)}$$
where $f_{ij}$ indicate the last-layer features of the matching score for the detected object pair in the respective frames *t* and t−1. $N_{t-1}$ is the total number of the detected objects in the frame t−1.
(8)$$\begin{array}{cc} \; & {s_{j}}^{start}=\frac{1}{1+\exp\left(-{f_{j}}^{t-1}\right)}\\ \; & {s_{i}}^{end}=\frac{1}{1+\exp\left(-{f_{i}}^{t}\right)} \end{array}$$
where ${f_{j}}^{t-1},{f_{i}}^{t}$ denote the last-layer features of the detected objects in frame t−1 and *t*, respectively.

We finally aggregate all the prediction scores to a new vector $S=[{s_{i}}^{det},{{s_{i}}_{j}}^{A},{s_{j}}^{start},{s_{i}}^{end}]$ for the optimization of the data association problem.

Considering the graph structure for all the detected objects, the data association problem can be formulated as the min-cost flow graph problem [2,27]. Specifically, we use these obtained prediction scores to define linear constraints, and then find an optimal solution for the matching problem.

There are two circumstances for a certain true positive object in previous frame t−1. It can be matched to another object in current frame *t*, or it starts a new trajectory. As a result, we define the linear constraint as Equation (Equation 9).
(9)$${s_{j}}^{det}={s_{j}}^{start}+\sum_{i=0}^{N}{{s_{i}}_{j}}^{A}$$

Similarly, a certain true positive object in current frame *t* can be matched to another object in previous frame t−1, or it ends a trajectory. Consequently, the linear constraint can be defined as Equation (Equation 10).
(10)$${s_{i}}^{det}={s_{i}}^{end}+\sum_{j=0}^{M}{{s_{i}}_{j}}^{A}$$

Finally, we formulate the data association problem as
(11)$$\begin{array}{cc} \; & \underset{s}{\arg\min}=\Theta {\left(X\right)}^{T}S\\ \; & s.t.\;\mathrm{CS}=0,S\;\in \;\left\{0,1\right\} \end{array}$$
where Θ(X) indicates a flattened vector that comprises all the prediction scores. CS is a matrix form that satisfies two linear constraint Equations (Equation 9) and (Equation 10).

## 4. Experiments

### 4.1. Dataset

We first evaluate our neural network on the KITTI object tracking benchmark [43,44]. The benchmark consists of 21 training sequences and 29 testing sequences. We split the training sequences into 10 sequences for training and 11 sequences for validation. As a result, we obtain 3975 frames for training and 3945 frames for validation. The dataset is captured from a car equipped with two color/gray stereo cameras, one Velodyne HDL-64E rotating 3D laser scanner and one GPS/IMU navigation system. Each object is annotated with a unique identity number (ID) across the frames in the sequences, 2D bounding boxes 3D bounding boxes parameters. We measure the distance between the predicted bounding box and corresponding bounding box of matched object-hypothesis by calculating the intersection over union (IoU).

### 4.2. Evaluation Metrics

The evaluation metrics to assess the performance of tracking methods are based on CLEAR MOT [45,46]. Specifically, MOT precision (MOTP) measures the average total error of distances for all the frames in the sequences as defined in Equation (Equation 12), and it indicates the total misalignment between the predicted bounding boxes and corresponding matched object-hypothesis.
(12)$$MOTP=\frac{\sum_{i,t}{d_{t}}^{i}}{\sum_{t}c_{t}}$$
where ${d_{t}}^{i}$ indicates the distance between the bounding box of *i*-th object and corresponding matched hypothesis in the frame *t*. $c_{t}$ indicates the total number of the matched objects in the frame *t*.

MOT accuracy (MOTA) measures the total tracking accuracy for all the frames as defined in Equation (Equation 13).
(13)$$MOTA=1-\frac{\sum_{t}({FN}_{t}+{FP}_{t}+{IDSW}_{t})}{\sum_{t}{GT}_{t}}$$
where ${FN}_{t},{FP}_{t},{IDSW}_{t},{GT}_{t}$ denote the total number of false negative objects, false positive objects, identity switches, and ground truth, respectively, in the frame *t*.

Besides, the work in [46] introduces other metrics to improve the tracking assessment, such as mostly tracking (MT) indicating the percentage of the entire trajectories in the sequences that could cover more than 80% in total length, mostly lost (ML) indicating the percentage of the entire trajectories in the sequences that could cover less than 20% in total length, and partial tracked (PT) indicating 1−MT−ML.

### 4.3. Training Settings

Our model is implemented using Pytorch v1.1. We trained our model on a ThinkStation P920 workstation with one NVIDIA GTX 1080Ti, and use Adam as a training optimization strategy with the initial learning rate 3 × 10^−4^. Besides, the super convergence training strategy is employed to boost the training processing and the maximum learning rate is set to 6 × 10^−4^.

### 4.4. Results

Table 1 shows that our Relation3D model achieves competitive results compared to recent state-of-the-art online tracking methods. Compared to the 2D tracking methods, our model is just slightly behind *MASS* [47], but outperforms all other 2D trackers. However, we discuss that the 2D information in our network still provides main contributions, as we observe the difference between our results and other 2D trackers is not quite much. We analyze the reason that the 3D information in our model also comes from the 2D detector *RRC-Net* [38]. As a result, we believe that inefficient 2D results are likely to lead to worse tracking performance.

Compared to the 3D tracking methods, our model achieves the best performance. However, Kalman filter method *AB3DMOT* [4] could perform in real-time and achieve the best ID−SW and Frag performance. Due to the fact the occlusion condition in our model is not taken into account, our ID−SW and Frag results are not efficient.

Compared to the 2D&3D tracking methods, it is easy to observe that our MOTA and MOTP results outperform those of *GNN3DMOT* [14], whose feature interaction mechanism employs an MLP network to exploit the discriminative features. It shows the effectiveness of our RelationConv operation for feature interaction. Besides, *GNN3DMOT* uses the 3D detections to obtain the 2D information by projecting the point cloud to the image, which leads to only 80.40% MOTA. Consequently, we believe that employing state-of-the-art 2D detectors directly is more beneficial for the 3D trackers. Compared to *mmMOT* [2], our model is slightly better as it is beneficial from our RelationConv operation and the combination of both the appearance features and motion features.

Table 2 indicates the effectiveness of different fusion methods. It shows that the concatenation operation outperforms the addition and attention-based weighted sum methods. We believe that the concatenation operation is capable of better exploiting the information from the features obtained from different sensors. Specifically, the addition operation is inefficient to align the features captured from 2D RGB images and 3D point clouds, making it difficult to learn discriminative feature after the element-wise addition. Although the attention-based weighted sum method enables highlightng the importance for the different features, the concatenation operation is a more general operation that gathers all the information from different modalities.

Table 3 shows that the performance is best when the edge feature ${{y_{i}}_{j}}^{t}=\left|{x_{i}}^{t}-{x_{j}}^{t-1}\right|$. We observe that the first option ${x_{j}}^{t-1}$ only considers the neighborhood information without involving the center nodes. The third option ${x_{i}}^{t}-{x_{j}}^{t-1}$ uses the relative distance value between the object pair as the edge feature, which is efficient when dealing with spatial data (e.g., point cloud). However, the result represents that the absolute value operator is more efficient when we learn discriminative features for the similarity. One possible reason is that only the absolute distance of the two high-level features is needed to measure the discrimination of different objects, which is beneficial for the model robustness. The fourth option $\left[{x_{i}}^{t},{x_{j}}^{t-1}\right]$ encodes the edge feature by combining the individual information without explicitly considering the distance between each pair of objects.

We also investigate the effectiveness of the appearance feature, motion feature, and different convolutional operations as shown in Table 4, which indicates that the motion feature could contribute the MOTA by 0.35%, and significantly reduce the number of ID−SW and Frag by 39 and 30, respectively, which convincingly verifies that the motion cues could help to match the correct objects for data association. Besides, our RelationConv outperforms the MLP method by 0.24% for MOTA metric and also performs much better in terms of ID−SW and Frag metrics.

It is worth highlighting that the MOTP metric is only related to the distance between predicted bounding boxes and corresponding matched object-hypothesis as shown in Equation (Equation 12). Besides, the MT and ML are irrelevant to the measurement of whether the IDs of the objects remain the same throughout the entire sequence. As a result, the metrics of MOTP,MT,ML are unlikely to be updated during our ablation investigation (see Table 2, Table 3 and Table 4), as long as the results of object detection performance remain the same.

We also study the effectiveness of different nonlinear activation functions in Equation (Equation 5) for our RelationConv operation. As shown in Table 5, the functions RELU and LeakyRELU achieve competitive results, which indicates that they are more likely to regress the model to the optimal solution. However, the sigmoid function performs the worst. We discuss the reason that its disadvantages, such as information loss due to vanishing gradients, make the model difficult to achieve optimization.

We finally provide some examples of the tracking results as shown in Figure 5, which includes three consecutive frames. All the detected objects are included in the green oriented 3D bounding boxes. We assign different IDs to different objects in each frame, and then track them by assigning the same IDs for the same objects in the different frames.

### 4.5. Ablation Study

We investigate different hyperparameter settings to evaluate the effectiveness of our model on the KITTI object tracking benchmark [43].

## 5. Conclusions

We proposed a deep affinity network, named Relation3DMOT, able to learn discriminative features and associate the objects in adjacent frames for 3D MOT. We employed a joint feature extractor to capture the 2D/3D appearance feature and 2D motion feature from 2D images and 3D point clouds, respectively, followed by a feature interaction module able to enhance the feature communication among objects in different frames. We also proposed a novel and efficient convolutional operation, named RelationConv, to abstract semantic and contextual relations for each object pair. Finally, we carried out extensive experiments on the KITTI object tracking benchmark to demonstrate the effectiveness of our Relation3DMOT tracker.

In the future, we plan to improve our model by considering objects with hard occlusion between frames. Further to this, it would be interesting to develop an end-to-end framework for the joint task of object detection and tracking, as we believe that it could avoid the decoupling issues that might arise when we deal with the object detection and tracking separately.

## Figures and Tables

**Figure 1 sensors-21-02113-f001:**
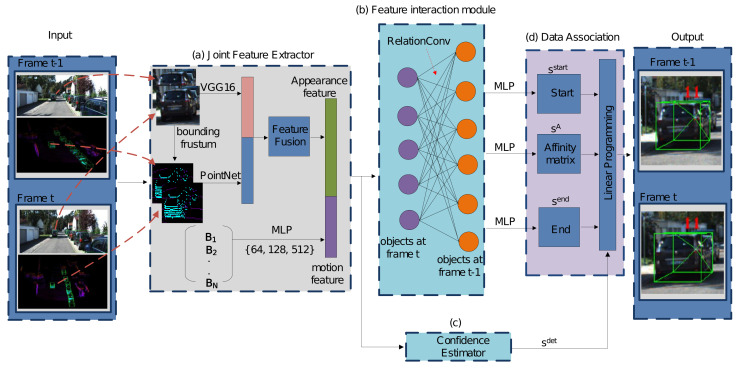
Relation3DMOT architecture. The architecture consists of 4 main parts. The *Joint Feature extractor* (**a**) takes the objects with 2D/3D bounding boxes as input, and then applies 2D feature extractor VGG-16 and 3D feature extractor PointNet on the cropped image and point cloud, respectively, to capture 2D/3D features, followed by a feature fusion module to obtain the appearance feature for each detected object. The motion feature is extracted by applying the 2D bounding boxes on a proposed MLP network. Note that the numbers in the brace indicate the number of the learning convolutional parameters for the MLP network. Finally, we achieve fusion by concatenating the appearance feature and the motion feature for each object. $\left\{B_{1},B_{2},\ldots B_{N}\right\}$ denotes a list of the 2D bounding boxes information. In the *Feature Interaction module* (**b**), we propose the novel RelationConv operation to build a feature interaction module for discriminative feature extraction. The *Confidence Estimator* (**c**) predicts whether the detected objects are true positive. In the *Data Association module* (**d**), we finally learn an affinity matrix and predict several binary scores to optimize the data association problem.

**Figure 2 sensors-21-02113-f002:**
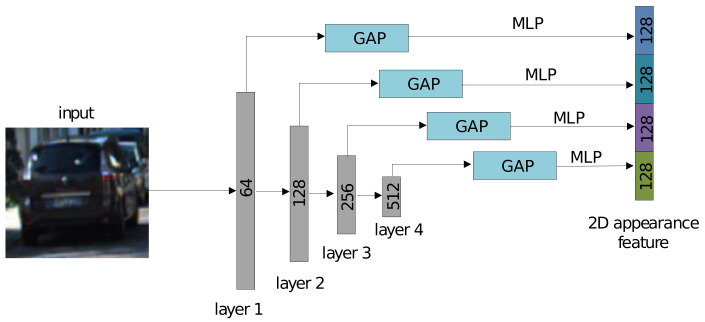
Modified VGG-16. The module takes the cropped and resized object patches with dimension 224×224 as input, and the layers output 64, 128, 256, 512 channels of the features, respectively, after a max pooling operation. Next, the global average pooling (GAP) operation and multi-layer perceptron (MLP) are used to obtain semantic features with 128 channels for each layer. Finally, all the layers are concatenated together to generate the global feature that contains the information in all the layers.

**Figure 3 sensors-21-02113-f003:**
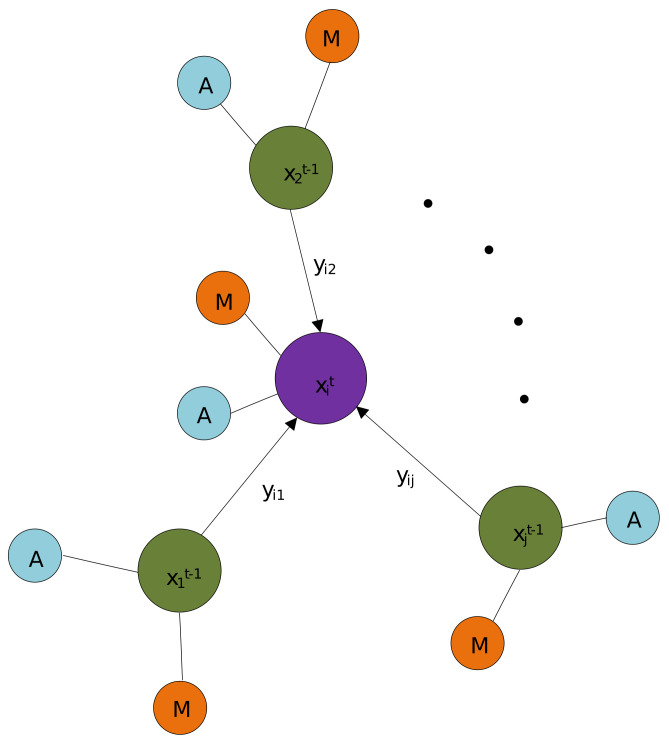
Graph construction. A certain detected object in current frame *t* is represented as $x_{i}^{t}$. The objects in previous frame t−1 are then defined as $x_{j}^{t-1}$, which are also treated as the neighborhood of the Node $x_{i}^{t}$. *A* and *M* are the appearance feature and the motion feature for respective nodes. $y_{ij}$ is the edge feature connecting *i*-th object node in current frame *t* and *j*-th object node in previous frame t−1.

**Figure 4 sensors-21-02113-f004:**
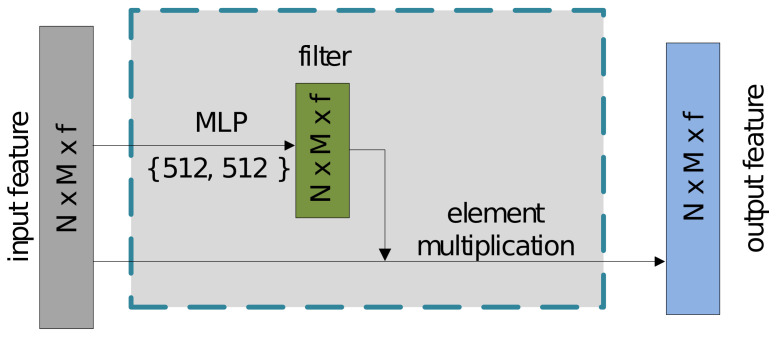
RelationConv operation structure. Assume that frame *t* and frame t−1 include *N* and *M* detected objects (nodes), respectively. The proposed RelationConv operation takes N×M nodes with *f*-dimension features as input. An MLP subnetwork is then employed to learn a flexible filter, which applies an element multiplication operation over the input feature to obtain the semantic output. Note that the numbers in the brace indicate the number of the learning convolutional parameters in the MLP subnetwork.

**Figure 5 sensors-21-02113-f005:**
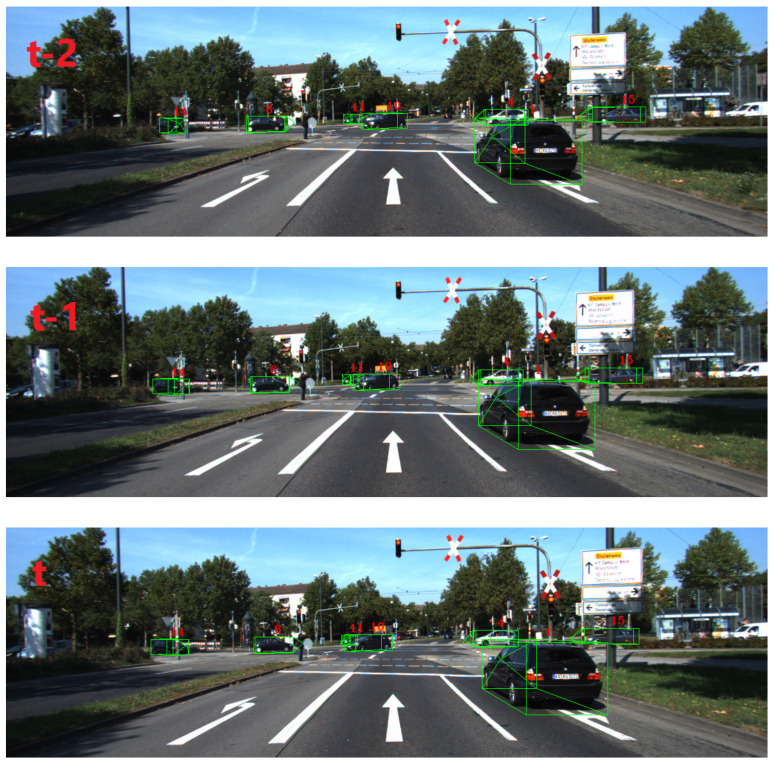
Qualitative results on the KITTI object tracking benchmark validation dataset.

**Table 1 sensors-21-02113-t001:** 3D tracking results on KITTI tracking test set.

Methods	Input	MOTA (%)	MOTP (%)	ID-SW	Frag	MT	ML
BeyondPixels [48]	2D	84.24	**85.73**	468	944	73.23	**2.77**
MASS [47]	2D	**85.04**	85.53	301	744	**74.31**	**2.77**
extraCK [49]	2D	79.99	82.46	343	938	62.15	5.54
JCSTD [50]	2D	80.57	81.81	61	643	56.77	7.38
IMMDP [51]	2D	83.04	82.74	172	365	60.62	11.38
AB3DMOT [4]	3D	83.84	85.24	**9**	**224**	66.92	11.38
PMBM [30]	3D	80.39	81.26	121	613	62.77	6.15
mono3DT [3]	3D	84.52	85.64	377	847	73.38	**2.77**
DSM [52]	2D + 3D	76.15	83.42	296	868	60.00	8.31
FANTrack [53]	2D + 3D	77.72	82.32	150	812	62.61	8.76
mmMOT [2]	2D + 3D	84.43	85.21	400	859	73.23	**2.77**
GNN3DMOT [14]	2D + 3D	80.40	85.05	113	265	70.77	11.08
OURS	2D + 3D	84.78	85.21	281	757	73.23	**2.77**

**Table 2 sensors-21-02113-t002:** Effectiveness of different fusion methods.

Fusion Method	MOTA (%)	MOTP (%)	ID-SW	Frag	MT (%)	ML (%)
add	91.72	90.35	106	210	90.28	0.9
concatenate	**92.33**	**90.35**	**38**	**143**	**90.28**	**0.9**
weighted sum	91.94	90.35	82	187	90.28	0.9

**Table 3 sensors-21-02113-t003:** Effectiveness of different edge feature aggregation operations.

Edge Feature	MOTA (%)	MOTP (%)	ID-SW	Frag	MT (%)	ML (%)
${x_{j}}^{t-1}$	90.94	90.35	193	290	90.28	0.9
$|{x_{i}}^{t}-{x_{j}}^{t-1}|$	**92.33**	**90.35**	**38**	**143**	**90.28**	**0.9**
${x_{i}}^{t}-{x_{j}}^{t-1}$	91.85	90.35	92	194	90.28	0.9
$\left[{x_{i}}^{t},{x_{j}}^{t-1}\right]$	91.96	90.35	80	181	90.28	0.9

**Table 4 sensors-21-02113-t004:** Effectiveness of appearance feature (A), motion feature (M), and different convolutional operation.

Feature	MOTA (%)	MOTP (%)	ID-SW	Frag	MT (%)	ML (%)
A + MLP	91.74	90.35	104	202	90.28	0.9
A + M + MLP	92.09	90.35	65	172	90.28	0.9
A + RelationConv	91.99	90.35	76	178	90.28	0.9
A + M + RelationConv	**92.33**	90.35	**38**	**143**	90.28	0.9

**Table 5 sensors-21-02113-t005:** Effectiveness of different activation functions for the proposed RelationConv operation.

Activation Function	MOTA (%)	MOTP (%)	ID-SW	Frag	MT (%)	ML (%)
RELU	**92.33**	90.35	**38**	**143**	90.28	0.9
Leaky RELU	91.92	90.35	84	190	90.28	0.9
Sigmoid	91.33	90.35	110	197	90.28	0.9

## Data Availability

Publicly available datasets were analyzed in this study. This data can be found here: http://www.cvlibs.net/datasets/kitti/eval_tracking.php (accessed on 16 March 2021).

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
