# Peer review of "Relation3DMOT: Exploiting Deep Affinity for 3D Multi-Object Tracking from View Aggregation"

_sensors, 2021, doi:10.3390/s21062113_

Round 1

Reviewer 1 Report

General Comments:

The paper presents an approach to track multiple objects in 3D space. The approach is based on joint feature extractor that fuses information from 2D images and 3D point clouds, then a convolutional operation to extract correlation between each pair of objects in the adjacent frames. The topic is important for autonomous navigation, and the proposed ideas are interesting.

Specific Comments:

  1. Sub-Section 3.3.3: Motion features should be clearly defined, and their extraction should be justified. Does this extraction take speed and blur into account?
  2. Sub-Section 3.3.4: The “weights” are not clearly defined in the weighted sum operation during the fusion process.
  3. Sections 3.4 and 3.5: The term “confidence” (or “affinity confidence”) is not clearly defined, despite its major role in data association.
  4. The use of ReLU in proposed Equation (3) should be justified (e.g., versus sigmoid or other functions).
  5. The role of M, N and finding their proper values remain unclear.

Reviewer 2 Report

The topic addressed in the paper is potentially interesting in both theory and practice.  In general, the paper is clear and the results are well stated and presented. The paper presents interesting results and comparisons that can be extended to experimental results and the method can be applied to other issues. However, there are some points that are not very clear and should be addressed in the revised version.

+ In the introduction, it is suggested that the novel index of this paper should be explained in detail. And the introduction should be added to do a better job of explaining the existing methods and why they are or are not valuable.

+ The reviewer also recommends that the authors need to add more results of different cases. This makes the readers easily realize the better performance of your method.

+ The explanations and analysis of the results should be enriched to show the validity of the data. Especially, Figure 4 was not mentioned and explained in the text of the paper.

Round 2

Reviewer 1 Report

The Authors have carefully addressed all of the Reviewers’ comments. The current version is clear and suitable for publication in MDPI Sensors.